# Cholesterol as an Endogenous Ligand of ERRα Promotes ERRα-Mediated Cellular Proliferation and Metabolic Target Gene Expression in Breast Cancer Cells

**DOI:** 10.3390/cells9081765

**Published:** 2020-07-23

**Authors:** Faegheh Ghanbari, Sylvie Mader, Anie Philip

**Affiliations:** 1Division of Plastic Surgery, Department of Surgery, Faculty of Medicine, McGill University, Montreal, QC H3G 1A4, Canada; faegheh.ghanbaridivshali@mail.mcgill.ca; 2Department of Biochemistry and Molecular Medicine, Institute for Research in Immunology and Cancer, Université de Montréal, Montréal, QC H3C 1J7, Canada; sylvie.mader@umontreal.ca

**Keywords:** breast cancer, cholesterol, estrogen related receptor α, statins, human pregnancy serum

## Abstract

Breast cancer is the 2nd leading cause of cancer-related death among women. Increased risk of breast cancer has been associated with high dietary cholesterol intake. However, the underlying mechanisms are not known. The nuclear receptor, estrogen-related receptor alpha (ERRα), plays an important role in breast cancer cell metabolism, and its overexpression has been linked to poor survival. Here we identified cholesterol as an endogenous ligand of ERRα by purification from human pregnancy serum using a GST-ERRα affinity column and liquid chromatography-tandem mass spectrometry (LC-MS/MS). We show that cholesterol interacts with ERRα and induces its transcriptional activity in estrogen receptor positive (ER+) and triple negative breast cancer (TNBC) cells. In addition, we show that cholesterol enhances ERRα-PGC-1α interaction, induces ERRα expression itself, augments several metabolic target genes of ERRα, and increases cell proliferation and migration in both ER+ and TNBC cells. Furthermore, the stimulatory effect of cholesterol on metabolic gene expression, cell proliferation, and migration requires the ERRα pathway. These findings provide a mechanistic explanation for the increased breast cancer risk associated with high dietary cholesterol and possibly the pro-survival effect of statins in breast cancer patients, highlighting the clinical relevance of lowering cholesterol levels in breast cancer patients overexpressing ERRα.

## 1. Introduction

Breast cancer is the most frequently diagnosed, and second deadliest cancer in women, with more than 200,000 new patients, and approximately 40,000 estimated deaths per year only in the United States [1]. Therefore, it is necessary to reduce incidence and improve outcome by therapeutic approaches addressing known breast cancer risk factors. Obesity, dyslipidemia and high dietary cholesterol intake are critical risk factors for breast cancer in pre- and post-menopausal women [2]. Several studies have indicated that obesity is associated with a higher risk of breast cancer in both triple negative breast cancer (TNBC) [3,4], and in ER positive women [5,6]. Indeed, post-menopausal women with high dietary cholesterol intake have been reported to have a ~50% increase in the risk of breast cancer [2,7]. In addition, in several different mouse models of breast cancer, high dietary cholesterol alone resulted in a significant decrease in tumor latency, and an increase in tumor volume and total tumor burden [8,9,10,11,12]. Interestingly, it was shown that established breast cancer is associated with higher low-density lipoprotein (LDL), and very low-density lipoprotein (VLDL)-cholesterol, however no link was identified with total cholesterol or high-density lipoprotein (HDL) [13].

There is some evidence that high-cholesterol diet affects the biophysical properties of lipid raft microdomains of the plasma membrane and enhances signaling activity via phosphoinositide 3-kinase (PI3K) and AKT/protein kinase B in breast cancer cells [9]. However, blood cholesterol levels in the mouse model used above were far higher than in human hypercholesterolemia. In addition, in the above study, exogenous cholesterol concentrations required for cancer cell proliferation were much lower (nanomolar range) than those required for lipid raft formation. Thus, it is unlikely that the pathological effects of cholesterol in breast cancer progression occur via alterations in lipid raft structure and associated signaling pathways [14], raising the possibility that cholesterol functions as a signaling molecule in breast cancer cells.

Interestingly, some studies have shown that the cholesterol metabolite 27-hydroxycholesterol acts as a signaling molecule through ER and liver X receptor (LXR) in ER+ breast cancer cells, which may explain how hypercholesteremia increases the risk in ER+ breast cancer cells [2,11,15]. However, several studies have reported that obesity and high cholesterol intake increase the risk not only in ER+ breast cancer, but also in triple negative breast cancer [3,4,5,16], supporting the notion that cholesterol itself acts as a signaling molecule and that such signaling may involve pathways other than the ER pathway.

The nuclear receptor estrogen-related receptor alpha (ERRα; NR3B1) plays important roles in energy metabolism by regulating the expression of genes involved in cellular energy metabolism, including those encoding enzymes in the oxidative phosphorylation (OXPHOS), tricarboxylic acid (TCA) cycle, glycolysis and in anabolic biosynthesis pathways like lipid, amino-acid and nucleic acid biosynthesis [17,18,19]. Importantly, ERRα adapts the metabolic pathways to fuel tumor growth via its interaction with the peroxisome proliferator-activated receptor coactivator-1α (PGC-1α ) [20,21].

ERRα belongs to the family of orphan nuclear receptors for which no endogenous ligands have been identified [22]. ERRα levels have been shown to be upregulated in ER+ and, in particular, in TNBC cells and its overexpression is linked to poor survival in those patients [17,18,23,24]. Following the identification of ERRα, it was initially thought that ERRα and ERα may have a large overlap in target genes and activity and therefore, play similar roles in breast cancer. However, it was subsequently shown that only few genes that are commonly regulated by both ERRα and ERα in MCF7 breast cancer cells [18]. Consistently, ChIP-on-chip studies demonstrated that only approximately 18% of ERα target genes are co-regulated by ERRα in MCF7 cells [21]. Furthermore, despite a high degree of amino acid similarity (68%) in the DNA binding domains (DBDs) of ERRs and ERα, ERRs do not bind strongly to perfect palindromic ER response elements [19,21,25]. In addition, several studies have shown that pharmacological modulation of ERRα activity with inverse agonists decreases the proliferation of both ER-positive and -negative breast cancer cells in vitro as well as tumorigenicity in nude mice [17,26,27,28]. Taken together, the above findings indicate that ERRα exhibits ER-independent pro-tumorigenic activities in breast cancer cells.

Recently, we showed that estradienolone (ED), an endogenous steroid from human pregnancy, acts as an inverse agonist of ERRs [28], and during ED’s characterization, we identified cholesterol as an agonist of ERRα, a finding consistent with a recent report that demonstrated that cholesterol isolated from mouse brain or kidney acts as an endogenous agonist of ERRα [29]. To better understand the mechanism by which high cholesterol levels increase breast cancer risk, we examined in the present study whether cholesterol acts through the ERRα pathway in TNBC and ER+ breast cancer cells. We show that the potent effects of cholesterol on cellular responses, gene expression in TNBC and ER+ cells are mediated via the ERRα pathway.

## 2. Methods

### 2.1. Extraction and Identification of ERRα Endogenous Ligands

Human pregnancy blood samples were collected from healthy pregnant women at 28–38 weeks of gestation using the informed constant. The samples were centrifuged at 3000 rpm for 10 min at 4 °C and the plasma samples were pooled and stored at −20 °C for further analysis. In order to identify endogenous ligand(s) of ERRα, a GST-ERRα pull down assay was performed. GST-ERRα-LBD (ligand binding domain) plasmid was constructed and GST-ERRα-beads were produced as previously described [28]. Sample preparation involves removing proteins from human pregnancy serum using methanol (Sigma Aldrich, Mississauga, ON, Canada). The precipitated proteins were centrifuged and the supernatant was dried down under nitrogen and reconstituted in methanol (Sigma Aldrich). Equal amounts of GST-ERRα-beads and GST-beads were incubated with extracted samples in PBS for 24 h at 4 °C with continuous shaking. The supernatants, which contain the unbound steroids and lipids, were collected and the ERRα-ligand complexes were eluted using 10 mM L-glutathione reduced solution (Sigma Aldrich). The proteins were removed from the eluted samples as described above. The eluted samples were subjected to liquid chromatography tandem mass spectrometry (LC-MS/MS) analysis (model 1260 Infinity with 1260 Infinity Diode Array Detector HS, Agilent Technologies, Santa Clara, CA, USA, coupled with an Impact HD MS detector, Bruker, Milton, ON, Canada). An Agilent Zorbax Eclipse Plus C18 column (4.6 × 10 mm, 3.5 μm) was used for separation. Mobile phase A was methanol/water/0.1% formic acid (3:1, *v*/*v*), and mobile phase B was 100% isopropanol/0.1% formic acid (Millipore, Sigma-Aldrich). The elution gradient was held at 40% B for the first 0.5 min, 40–90% B from 0.5 to 4.5 min, held at 90% B from 4.5 to 6.5 min, 90–40% B from 6.5 to 6.6 min, held at 40% B from 6.6 to 7 min. The flow rate of the mobile phase was 1 mL/min, and the injection volume was 20 μL. For mass spectrometry, the electrospray ionization was operated in positive and total scan mode. For mass spectrometry, capillary voltage was 4000 V, fragmentor voltage was 500 V; nebulizer gas was 73 psi; drying gas temp was 350 °C with the flow of 12 L/min; *m*/*z* range was from 150 to 800 Daltons.

### 2.2. GST-ERRα Pull Down Assay

To confirm that cholesterol directly binds to ERRα-LBD, a GST-ERRα pull down assay was used as described above. Briefly, 2 μM of cholesterol, XCT-790 or estradiol (E2) (Sigma Aldrich) were incubated with beads-GST-ERRα-LBD and beads-GST. The pull down and supernatants were dissolved in methanol, as described earlier. Cholesterol concentrations were measured using a multiple reaction monitoring (MRM) mode by LC-MS/MS as above. However, XCT-790 and E2 concentrations were determined using a UV–vis spectrophotometer (Cary Series UV–vis-NIR spectrophotometer, Agilent Technologies) at the maximum wavelength of 368 nm and 281 nm, respectively.

### 2.3. Tryptophan Fluorescence Quenching Assay

GST-ERRα-LBD (PV4665) was purchased from Life Technologies (Grand Island, NY, USA). Four hundred nM of GST-ERRα-LBD was incubated with varying concentration of cholesterol, XCT-790, and E2 as previously described [28]. Fluorescence excitation was at 295 nm and the florescent emission was measured at 310 nm using a microplate reader (Infinite M200PRO, TECAN, Männedorf, Switzerland). The dissociation constant (Kd) was determined using Graph Pad software (San Diego, CA, USA).

### 2.4. Cell Culture

Human embryonic kidney 293 (HEK-293) cells were purchased from Sigma. The MDA-MB-231 and MCF-7 cell lines were purchased from ATCC (Manassas, Virginia, USA). All the above-mentioned cell lines were cultured in DMEM supplemented with 10% FBS and 1% penicillin/streptomycin. For all experiments, cells were switched 24 h before cell treatments to fresh phenol red free medium (21063029, Thermo Fisher Scientific, Grand Island, NY, USA) supplemented with 2% lipoprotein depleted and charcoal-stripped FBS. Lipoprotein depleted FBS was purchased from Kalen Biomedical LLC (Germantown, MD, USA) and was charcoal-stripped in order to remove steroid hormones as described previously [30]. Lovastatin (sc-200850A, Santa Cruz Biotechnology, Santa Cruz, CA, USA), a known cholesterol lowering drug, was used to decrease cholesterol intracellular level. XCT-790 and compound 29 (cpd29), known synthetic inverse agonists of ERRα, were used to decrease ERRα transcriptional activity. XCT-790 (X4753-5MG) was purchased from Sigma Aldrich, and cpd29 was a generous gift from Dr. Donald McDonnell (Duke University, Durham, NC, USA).

### 2.5. Antibodies

Rabbit monoclonal anti-ERRα antibody (ab76228), mouse monoclonal anti-VEGF antibody [VG-1] (ab1316), and mouse monoclonal anti-alpha tubulin antibody (ab7291) were from Abcam (Cambridge, MA, USA). Anti-PGC-1α mouse (4C1.3. mAb) antibody, and mouse monoclonal anti-ERRα antibody (sc-65715) were from Millipore Sigma and Santa Cruz Biotechnology, respectively. Anti-GAPDH rabbit (mAB#2118) was purchased from Cell Signaling Technology (Danvers, MA, USA).

### 2.6. Luciferase Reporter Assay to Determine Cholesterol’s Effect on ERRα’s Transcriptional Activity

To determine whether cholesterol regulates transcriptional activity of ERRα, HEK-293 were transfected with the pS2-Luc reporter plasmid (400 ng), with or without ERRα expression plasmid (300 ng), with or without the proliferator-activated receptor gamma coactivator-1 (PGC-1α) co-activator expression plasmid (300 ng), and a Renilla luciferase expression vector (20 ng) as previously described [31]. 48 h after transfection, cells were treated with varying concentrations of cholesterol and XCT-790 (5 μM) as a positive control. Luciferase activity was measured after 24 h and values were normalized to Renilla. The values shown are representative of three independent experiments.

### 2.7. Immunoblotting and Immunoprecipitation

MDA-MB-231 cells were seeded in 10 cm plates and were treated with vehicle (veh), lovastatin (lova), cholesterol + lovastatin (chol + lova) or cholesterol (chol), all at 5 μM. After 24 h the cells were harvested and lysed with non-denaturing lysis buffer (20 mM Tris-HCl, pH 8, 137 mM NaCl, 10% glycerol, 1% Nonidet P-40, 2 mM EDTA, and protease inhibitors (Sigma Aldrich)). Co-immunoprecipitation (co-IP) was carried out as described previously [32]. The above-mentioned cell lysates (500 μg of total protein) were immunoprecipitated with rabbit anti-ERRα antibody (10 μg) or control rabbit immunoglobulin G (IgG, 10 μg) (12-370(CH), Millipore) overnight at 4 °C with end over end shaking, followed by a 2 h incubation with Protein A Sepharose 4B (20 μL) (10-1141, Invitrogen) at 4 °C. Supernatants were removed after sample centrifugation. The pellets containing beads were washed three times with ice-cold lysis buffer and bead-bound proteins were eluted, denatured and immunoblotted using mouse anti-ERRα antibody or mouse anti-PGC-1α antibody. 2% of the total cell lysates (TCL) were used to detect endogenous levels of ERRα and PGC-1α in cells treated with vehicle, lovastatin, cholesterol + lovastatin or cholesterol.

For MCF-7 cells, the immunoprecipitation procedure was slightly different. Cells were seeded in 10 cm plate and treated with vehicle or 10 μM cholesterol. After 24 h of treatment, a Pierce co-immunoprecipitation (Co-IP) kit (26149, Thermo Fisher Scientific) was used for the preparation of whole cell lysates and immunoprecipitation using an anti-ERRα antibody, according to the manufacturer’s instructions. Briefly, the anti-ERRα antibody was first covalently immobilized to AminoLink Plus coupling resin for 2 h. The resin was then washed and incubated with 500 μg of the above-mentioned cell lysates overnight. After incubation, the resin was washed and the protein complexes were eluted. A negative control (Pierce Control Agarose Resin), provided with the IP kit to determine nonspecific binding, received the same treatment as the co-IP samples, including the anti-ERRα antibody. In this case, the coupling resin is not amine-reactive, therefore, it prevents the antibody from covalent immobilization onto the resin. The eluted co-IP proteins were analyzed by SDS-PAGE and immunoblotted (IB) with a PGC-1α or ERRα antibody. 2% of the total cell lysates (TCL) were used to detect endogenous levels of ERRα and PGC-1α in cells treated with vehicle or cholesterol.

### 2.8. siRNA Transfection

siRNAs directed against ERRα (AM16708/289481, Invitrogen, Carlsbad, CA, USA) with the sense sequence 5′-CCGCUUUUGGUUUUAACC-3′ and antisense sequence 5′-GGUUUAAAACCAAAAGCGG-3′ or control scrambled siRNAs (AM4611, negative control, Invitrogen) were transfected into MCF-7 and MDA-MB-231 cells using Lipofectamine RNAiMAX Transfection Reagent (Invitrogen) according to manufacturer’s instructions. At 48 h post-transfection, fresh phenol red-free medium containing 2% lipoprotein-depleted serum was added and cells were treated with cholesterol and/or lovastatin (5 μM for MDA-MB-231, 10 μM for MCF-7 cells). The knock-down breast cancer cells were used for cell growth assays, immunoblotting and qPCR analyses.

### 2.9. RNA Preparation and Analysis

Total RNA was extracted using an RNeasy mini kit (74104, Qiagen, Germantown, MD, USA). One microgram of total RNA was used for the first-strand synthesis with high-capacity cDNA reverse transcription kit (4368814, Life Technologies, Grand Island, NY, US). Real-time PCR was performed using EvaGreen qPCR master mix (Applied Biological Materials Inc., Mastermix-R, Diamond, Richmond, BC, Canada) with gene-specific primers. The targets included in this study are: isocitrate dehydrogenase 3A (IDH3A), pyruvate dehydrogenase kinase 4 (PDK4), vascular endothelial growth factor (VEGF), glutathione S-transferase M1 (GSTM1), superoxide dismutase 2 (SOD2) and secreted phosphoprotein 1(SPP1). The sequences of the above-mentioned primers are indicated in Appendix A. Real-time PCR was performed on the 7500 real-time PCR system (Applied Biosystems, Woburn, MA, USA). Data analysis was performed using real-time PCR software 7500, version 2.1 (Applied Biosystems). The relative RNA concentrations of the genes of interest were determined using the comparative threshold cycle (Δ*C_T_*) method after normalization to the endogenous control glyceraldehyde 3-phosphate dehydrogenase (GAPDH). 

### 2.10. Immunocytochemistry (ICC)

MCF-7 and MDA-MB-231 cells were cultured on cover slips (170-C12MM, Ultident Scientific, Saint-Laurent, QC, Canada). The cells were treated with vehicle or cholesterol (5 μM) for 24 h, and then washed with PBS twice and fixed with 4% paraformaldehyde in PBS for 10 min, followed by three PBS washes. The cells were blocked using 3% BSA for 30 min, and then incubated with the VEGF primary antibody overnight at 4 °C. Following three washes the cover slips were incubated with the secondary antibody (Alexa Fluor 488 anti-mouse, A11029, Molecular Probes, Eugene, OR, USA) for 1 h. The cover slips were washed again and the nuclei were counterstained with DAPI, and cover slips were attached. The images were obtained using a florescent microscopy (IX71, Olympus, Richmond Hill, ON, Canada) with a 20× objective and a LSM780 laser scanning confocal microscope (Zeiss, White Plains, NY, USA) with a 20×/0.4 LD “Plan-Neofluar” objective.

### 2.11. Cell Proliferation Assay

In order to obtain half maximal effective concentrations (EC_50s_) of cholesterol in MDA-MB-231 and MCF-7 cells, and determine the half maximal inhibitory concentration (IC50) of lovastatin on MDA-MB-231 and MCF-7 cells, proliferation assays with an MTS Cell Proliferation Assay kit (G3582, Promega, Madison, WI, USA) were performed. The MTS assay is based on the reduction of the MTS tetrazolium compound by viable cells to generate a colored formazan dye in metabolically-active cells. According to manufacturer’s instructions, MCF-7 and MDA-MB-231 cells were plated at a density of l0^4^ cells per well in 96 well plates. Cells were treated with varying concentrations of cholesterol, lovastatin, lovastatin (5 μM) + cholesterol, and treatment medium was replaced every 48 h throughout the 5 days’ duration of the experiment. Also, MDA-MB-231 cells were treated with varying concentrations of compound 29, in the absence or presence of cholesterol (5 μM), and with medium replacement every 72 h throughout the 6 day duration of the experiment. MTS reagent (20 μL of per well) was then added to and was incubated for 2 h at 37 °C under standard culture conditions. The optical density (OD) value was measured at 490 nm using a microplate reader (Infinite M200PRO, TECAN).

### 2.12. Migration Assay

Scratch-wound migration assays were performed on MDA-MB-231 cells. Confluent monolayer cells were pre-incubated with serum-free and phenol red-free medium for 24 h to inhibit cell proliferation. Then monolayers of MDA-MB-231 cells were scratched using a 1mL pipet tip to create a cell-free line and were washed three times to remove cellular debris. The culture plates were replenished with fresh red phenol-free media containing vehicle, cholesterol, lovastatin and/or XCT-790 (all at 5 μM). Wound closure was monitored at times 0 and 24 h, and representative images were photographed using a bright-field microscope (Evos XL core, Life Technologies). Wound width for each treatment was calculated based on averaging six individual measurements at time point 0 and 24 h using ImageJ software. Cell migration was expressed as a percentage of the scratch area filled by migrating cells at 24 h post scratch: migration % = (scratch width at *T* 0 h − scratch width at *T* 24 h/width at *T* 0 h) × 100.

### 2.13. Statistical Analysis

All values are expressed as means of at least three independent experiments ± SEM. The statistical significance of differences between two experimental groups was analyzed by a two-tailed Student *t*-test, and comparisons between more than two groups were analyzed by one-way ANOVA. The experiments were repeated at least three times to obtain *p* values. * represents *p* < 0.05, and was considered to be statistically significant.

## 3. Results

### 3.1. Identification of Cholesterol as a Candidate Endogenous Ligand of ERRα

To identify endogenous ligands of ERRα from human pregnancy serum, steroids and lipids were extracted from samples in methanol and incubated with beads-GST-ERRα-LBD and beads-GST (as a negative control). The eluted samples were analyzed using LC-MS/MS in full scan mode with a mass range of 200–500 m/z. In order to identify specific binders of ERRα-LBD, the mass spectra obtained with beads-GST-ERRα-LBD and beads-GST were compared. As shown in Figure 1A, a fragment of 369.3 m/z was detected, which represents a daughter ion of cholesterol in the ESI system at the elution time of 3.2 min. The intensity of the cholesterol daughter ion (369.3 m/z) in beads-GST-ERRα-LBD is 5-fold higher than the one in beads-GST. This result suggests that cholesterol acts as an endogenous ligand of ERRα-LBD.

### 3.2. Cholesterol Directly Binds to ERRα and Increases Its Transcriptional Activity

In order to verify whether cholesterol directly binds to ERRα-LBD, we performed a GST-ERRα-LBD pull down assay (Figure 1B). As shown in Figure 1B, the concentration of cholesterol in the pull down fraction is approximately three times higher than in the supernatant fraction. As a reference, the concentration of XCT-790, a synthetic inverse agonist of ERRα, in the pulldown fraction is approximately 4 times higher than its concentration in supernatant. To demonstrate that cholesterol binding to ERRα-LBD is specific, we performed the same experiment using GST alone. In this negative control, cholesterol concentration is about 2.5 times lower in the pulldown than in the supernatant (Figure 1B). Together, these results suggest that cholesterol directly interacts with ERRα-LBD.

To quantify the relative affinity of cholesterol for ERRα, we performed tryptophan fluorescent quenching assays using GST-ERRα (Figure 1C). As indicated in Figure 1C, quenching of fluorescence increases in the presence of either cholesterol or XCT-790 (a positive control) in a dose-dependent manner, suggesting that cholesterol and XCT-790 bind to ERRα and change its conformation, resulting in changes in the fluorescent emission of the receptor in the presence of varying concentrations of cholesterol or XCT-790. However, in the presence of estradiol (E2) as a negative control, fluorescence quenching of the GST-ERRα protein remained unchanged. The K_d_ values for cholesterol and XCT-790 were determined at 213.4 and 49.94 nM, respectively.

We next determined the impact of cholesterol on the transcriptional activity of ERRα. HEK-293 cells were transiently transfected with a reporter plasmid (pS2Luc) and expression vectors for full length ERRα and/or the PGC-1α co-activator (Figure 1D). Cells were transfected with ERRα, PGC-1α or both, and then treated with varying concentrations of cholesterol; XCT-790 was used as a positive control. As demonstrated in Figure 1D, in the presence of both ERRα and PGC-1α, increasing concentrations of cholesterol significantly enhances ERRα transcriptional activity. In contrast, transfection with ERRα or PGC-1α alone does not significantly increase transcriptional activity. This indicates that the effects of cholesterol on ERRα transcriptional activity require both ERRα and PGC-1α. Taken together, these data demonstrate that cholesterol, as an endogenous ligand of ERRα, binds to ERRα-LBD with a relative affinity of 213.4 nM, and increases transcriptional activity of ERRα in a PGC-1α dependent manner.

### 3.3. Cholesterol Enhances ERRα-PGC-1α Interaction in Breast Cancer Cells

To determine whether cholesterol regulates ERRα-PGC-1α interaction in triple-negative (MDA-MB-231) and estrogen receptor-positive (MCF-7) breast cancer cells, co-immunoprecipitation experiments were performed. For MDA-MB-231 cells, the procedure involved treating the cells with vehicle, cholesterol, lovastatin, or lovastatin + cholesterol, and immunoprecipitating the cell lysates using an anti-ERRα antibody or control IgG. All samples were then subjected to immunoblotting with an anti-PGC-1α antibody. As demonstrated in Figure 2A, cholesterol significantly enhances the association of PGC-1α to ERRα compared to the vehicle. No significant decrease in ERRα-PGC-1α association was detectable in the presence of lovastatin, a cholesterol-lowering drug. Significantly, cholesterol was able to enhance ERRα-PGC-1α association even in the presence of lovastatin. Importantly, this association was not detectable when control IgG was used instead of ERRα antibody. Together, these data suggest that exogenous cholesterol increases ERRα and PGC-1α interaction in MDA-MB-231 cells.

The co-immunoprecipitation procedure for MCF-7- cells involved treating the cells with vehicle or cholesterol, and incubating the cell lysates with AminoLink Plus Coupling Resin and immunoprecipitating using anti-ERRα antibody. As shown in Figure 2B, cholesterol significantly increases the interaction of PGC-1α to ERRα. As expected, this interaction was not detectable in the negative control experiment using uncoupled Pierce Control Agarose Resin and anti-ERRα antibody (-ve ctl). Together, these results show that cholesterol treatment enhances the interaction of ERRα and PGC-1α in both MDA-MB-231 and MCF-7 cells.

### 3.4. Cholesterol Increases ERRα Levels in Breast Cancer Cells

To assess whether cholesterol regulates ERRα expression levels, we treated cells with varying concentrations of cholesterol, and ERRα protein and mRNA levels were measured. As shown in Figure 3A, in the presence of increasing concentrations of cholesterol in MDA-MB-231 cells, a significant increase in ERRα protein levels was observed. To determine whether cholesterol increases ERRα protein levels in the presence of lovastatin (a known lowering cholesterol drug), MDA-MB-231 cells were treated with vehicle, lovastatin, or lovastatin + cholesterol and subjected to immunoblotting (Figure 3B). As shown in Figure 3B, lovastatin does not alter ERRα protein levels. However, adding cholesterol in the presence of lovastatin significantly increases ERRα protein levels. Consistent with these results, we observed a significant induction in ERRα’s mRNA levels in MDA-MB-231 cells upon cholesterol treatment (Figure 3C). Moreover, as shown in Figure 3D, when MCF-7 cells were treated with increasing concentrations of cholesterol, ERRα protein levels were significantly increased. In agreement with these results, we observed a significant induction in ERRα mRNA level in MCF-7 cells upon cholesterol treatment (Figure 3E). Altogether, these data demonstrate that exogenous cholesterol significantly enhances the mRNA and protein levels of ERRα in MDA-MB-231 and MCF-7 cells.

### 3.5. Cholesterol Enhances ERRα-Induced Metabolic Target Genes Through ERRα Pathway

Next, we determined whether cholesterol regulates ERRα metabolic target genes in MDA-MB-231 and MCF-7 cells. The cells were first transfected with siRNA-control or siRNA-ERRα, followed by treatment with cholesterol. As shown in Figure 4A,B, upon cholesterol treatment, there is a significant increase in the expression of metabolic target genes of ERRα, including IDH3A, VEGF, PDK4, SOD2, GSTM1, and SPP1, in breast cancer cells.

However, in ERRα knockdown breast cancer cells, cholesterol does not enhance the expression of ERRα target genes in either MDA-MB-231 or MCF-7 cells. As shown in Figure 4A,B, the efficiency of knockdown-ERRα was 89.9% and 82.7% for MDA-MB-231 and MCF-7 cells, respectively. 

These data indicate that the induction of ERRα metabolic target gene expression by cholesterol is ERRα-dependent. To confirm that cholesterol increases expression of ERRα target genes in breast cancer cells via the ERRα pathway, levels of the VEGF protein were assessed in cells treated with siRNA-ERRα or siRNA-control in the presence or absence of cholesterol. As shown in Figure 4C–F, cholesterol increases VEGF protein expression in MDA-MB-231 and MCF-7 cells, as detected by immunoblotting and immunocytochemistry using anti-VEGF antibody. In the absence of ERRα, the stimulatory effects of cholesterol were abolished in both types of breast cancer cells. These data suggest that cholesterol enhances VEGF protein expression through ERRα.

### 3.6. Cholesterol Enhances Cellular Proliferation of Breast Cancer Cells via the ERRα Pathway

To determine whether cholesterol regulates cellular proliferation in MDA-MB-231 and MCF-7 cells, cells were treated with varying concentrations of cholesterol. As shown in Figure 5A–C, cholesterol enhances cellular proliferation of these cells in a dose-dependent manner, and the EC_50_s of cholesterol for MDA-MB-231 and MCF-7 cells are approximately 70 and 110 nM, respectively. These results showing that cholesterol enhances cell proliferation at low nano-molar concentrations support the notion that cholesterol may act as a signaling molecule in these cells. Interestingly, lovastatin decreases cell proliferation of breast cancer cells in a dose-dependent manner and cholesterol inhibits this effect (Figure 5D,E). As demonstrated in Figure 5F, the IC_50_ of lovastatin in MDA-MB-231 cells is 1.81 μM, which is slightly lower than the one in MCF-7 cells (5.34 μM), possibly due to the higher expression of ERRα in MDA-MB-231 compared to MCF-7 cells. 

To demonstrate that the effect of cholesterol on breast cancer cell proliferation is mediated via ERRα, the expression of ERRα was knocked down by siRNA and cells were treated with lovastatin or cholesterol (Figure 5G,H). As shown in Figure 5G,H, ERRα was successfully knocked down in MDA-MB-231 and MCF-7 cells and cholesterol-induced cell proliferation is abrogated when ERRα expression is suppressed. Similarly, lovastatin-induced inhibition of cell proliferation is abolished in ERRα-deficient cells. These results suggest that both cholesterol-induced cell proliferation, and lovastatin-induced inhibition of cell proliferation are mediated via ERRα. Consistent with the result shown in Figure 5D,E, cholesterol is able to rescue the lovastatin-induced inhibition of cell proliferation. The lovastatin inhibitory effect on cellular proliferation is likely due to lowering intercellular cholesterol level, although cholesterol-independent effects of lovastatin cannot be ruled out. In order to confirm that cholesterol mediates cell proliferation in an ERRα-dependent manner, we performed dose-competition assays between the ERRα antagonist cpd29 and cholesterol. As shown in Figure 5I (black bars), cpd29 decreases cell proliferation in a dose-dependent manner in MDA-MB-231cells. Importantly, cpd29 also decreases cholesterol induced cell proliferation in a dose dependent manner in those cells (Figure 5I gray bars). Collectively, these data show that cholesterol increases cell proliferation of both MDA-MB-231 and MCF-7 cells, acting via ERRα.

### 3.7. Cholesterol Rescues the Inhibitory Effect of Lovastatin on Cellular Migration, but not that of XCT-790 in MDA-MB-231 Cells

To verify whether the effects of cholesterol on breast cancer cell migration are mediated through ERRα, we performed a scratch assay. Cells were treated with lovastatin, cholesterol, and/or XCT-790. As shown in Figure 6A,B, adding exogenous cholesterol does not significantly increase cellular migration although a trend in that direction is observed upon cholesterol treatment for 24 h. It is possible that a significant increase in cell migration requires a cholesterol treatment duration of more than the 24 hours used in the current study (i.e., 48 h or 72 h). In order to further probe whether cholesterol displays any effect on breast cancer cellular migration within 24 h, we used lovastatin to decrease intracellular cholesterol levels. Interestingly, cholesterol is able to rescue the lovastatin-induced inhibition of cellular migration in a significant manner. Next, to verify whether ERRα mediates the stimulatory effect of cholesterol on breast cancer cellular migration in the presence of lovastatin, XCT-790, a small molecule inhibitor of ERRα activity, was used. As shown in Figure 6A, XCT-790 decreases MDA-MB-231 cellular migration. However, adding exogenous cholesterol was unable to rescue the migration inhibitory effect of XCT-790 even in the presence of lovastatin. Together, these results suggest that the inhibition of cell migration induced by the cholesterol lowering agent lovastatin is rescued by cholesterol. However, when ERRα is inhibited by XCT-790, cholesterol does not increase cell migration nor does it restore the effect of lovastatin.

## 4. Discussion

There is accumulating evidence that obesity and high blood cholesterol increase the risk of breast cancer recurrence [33,34,35], while the use of statins, known cholesterol lowering drugs, is linked to increased disease-free survival in breast cancer patients [11,35,36,37,38]. However, the underlying mechanisms by which high cholesterol levels increase breast cancer recurrence risk and mortality are poorly understood [35,38]. As ERRα orphan receptor is a master regulator of energy metabolism, and its levels are upregulated in breast cancer with overexpression associated with poor survival, we pursued identification of its endogenous ligands. We recently reported the identification of an estradienolone-like molecule (ED) from human pregnancy urine, as an endogenous inverse agonist of ERRα [28]. In the current study, we demonstrate that cholesterol isolated from human pregnancy blood acts as an endogenous agonist of ERRα. We show that cholesterol binds ERRα and enhances its transcriptional activity in ER-positive and triple-negative breast cancer cells, which overexpress ERRα. Furthermore, we demonstrate that cholesterol enhances the interaction of ERRα with its transcriptional co-activator, PGC-1α, resulting in the activation of several ERRα’s target genes (including VEGF and ERRα itself), and in promoting cellular proliferation and migration in an ERRα-dependent manner, in breast cancer cells. Importantly, lovastatin inhibits cell proliferation and migration in both ER positive and triple negative breast cancer cells, possibly due to a decrease in intracellular cholesterol levels [39], and cholesterol is able to rescue these effects of lovastatin. The anticancer effects of statins have been shown to involve multiple molecular pathways, including inhibition of protein kinase B (AKT)/mammalian target of rapamycin (mTOR) [40,41]. the Neverthless, the current study demonstrates thatin the presence of lovastatin, addition of cholesterol is able to restore the inhibitory effects of lovastatin on cell proliferation and migration of MDA-MB-231 cells via the ERRα pathway, while ERRα protein levels remain unchanged. This, together with our results showing that a well-characterized ERRα antagonist, cpd29 [42,43,44], is able to inhibit cholesterol-induced cellular proliferation supports the premise that cholesterol-induced cellular proliferation is mediated via ERRα.

Our finding that cholesterol isolated from human pregnancy blood is an endogenous agonist of ERRα is in agreement with the findings from another group using cholesterol isolated from mouse brain and kidney [29]. Our results showing that cholesterol binds directly and specifically to the purified ligand binding domain of ERRα, with a dissociation constant of approximately 210 nM, and increases transcriptional activity of ERRα in a PGC-1α-dependent manner in both ER-positive and triple-negative breast cancer cells, indicate that cholesterol acts as an endogenous agonist of ERRα-PGC-1α signaling in these cells. In addition, our findings suggest that the mechanism by which cholesterol enhances ERRα transcriptional activity involves increasing the recruitment of PGC-1α to ERRα, as detected by enhanced interaction between ERRα and PGC-1α in the presence of cholesterol, whether in the presence or absence of lovastatin. It is possible that cholesterol acts as an allosteric activator by binding to the ERRα protein and changes its conformation, leading to enhanced interaction with its coactivator PGC-1α, and thus promoting ERRα transcriptional activity.

The ERRα/PGC-1α/β complex is the main regulator of genes involved in energy metabolism and mitochondrial biogenesis and directs metabolic reprogramming in cancer cells. It has been reported that this complex controls the expression of genes involved in the TCA cycle, OXPHOS, lipid metabolism, and glycolysis in breast cancer cells [18,21]. It is thus significant that cholesterol binding to ERRα and cholesterol-mediated increase in ERRα-PGC-1α interaction result in increased expression of ERRα itself and its metabolic target genes including IDH3A, VEGF, SOD2, GSTM1, PDK4, SPP1 in breast cancer cells. The finding that the cholesterol-mediated increase in the expression of these genes requires ERRα in both ER-positive and triple-negative breast cancer cells provides a mechanistic explanation for the adverse effect of high circulating cholesterol levels and may explain the beneficial effect of statins on breast cancer outcome.

Cholesterol’s ability to increase ERRα mRNA and protein levels in a dose-dependent manner can be explained by ERRα specific auto-induction, as ERRα activates the promoter of its own gene, ESRRA, thus providing positive regulation of its own expression [18,21,27]. It is possible that when cholesterol binds to ERRα and enhances its interaction to PGC-1α, this leads to the binding of the ERRα/PGC-1α complex to the ESRRA promoter and induction of ERRα expression itself as well as that of the metabolic target genes of ERRα. IDH3A, a major metabolic target gene of ERRα, is a key enzyme in the TCA cycle, and is known to stimulate angiogenesis and metabolic reprogramming of cancer cells to provide the necessary nutrients for cancer cell growth [45,46]. Similarly, cholesterol-mediated increase in PDK4 is of significant interest in this regard, as it is also a key enzyme in glucose and fatty acid metabolism, and its expression is upregulated in breast cancer and correlates with poor patient outcomes [47]. Together, these findings suggest that cholesterol induces metabolic gene expression via its modulation of ERRα activity.

SOD2 and GSTM1 are responsible for the detoxification of reactive oxygen species (ROS) and electrophilic compounds, which are produced mainly by mitochondria in cancer cells [20,48]. The elevation of ROS was shown to be essential for the metabolic reprogramming toward glycolysis [49]. Based on our findings, we suggest that high cholesterol levels resulting in increased interaction of ERRα with PGC-1α and ERRα-PGC1α signaling provide protection against the production of ROS from oxidative stress by increasing cell detoxification enzymes like SOD2 and GSTM1, and thus help avoid irreversible damage on mitochondria and other organelles of cancer cells. In addition, as SPP1 is known to be a direct target gene of ERRα, and has been shown to be overexpressed in breast cancer cells and functionally contribute to cancer progression [48,49], our finding that the cholesterol-induced increase in its expression requires ERRα in breast cancer cells is consistent with cholesterol’s adverse effects on breast cancer outcome.

Vascularization is an important process in metastatic progression. ERRα and its coactivator PGC-1α have been reported to bind to the promoter of VEGF (known to be involved in tumor invasion and angiogenesis), and enhance its expression [18,50,51,52]. In the current study, we showed that the expression of VEGF is significantly increased in the presence of cholesterol in ER-positive and triple-negative breast cancer cells and that this cholesterol effect requires ERRα, strongly suggesting that cholesterol enhances ERRα-induced VEGF expression in breast cancer cells. The ability to proliferate and migrate are two metastatic hallmarks of cancer cells. Cholesterol promotes cell proliferation and migration in ER+ and triple-negative breast cancer cells in an ERRα dependent manner, whereas statin shows opposite effect. Importantly, cholesterol also rescues the effect of statin on proliferation and migration in an ERRα-dependent manner. Importantly, our results show that cholesterol increases cell proliferation of triple negative and ER+ breast cancer cells in a dose-dependent manner within a nanomolar range, implicating cholesterol as a signaling molecule. Lovastatin displays the opposite effect, decreasing both cell proliferation and migration of both cell types, and cholesterol rescues lovastatin’s effect. However, cholesterol is unable to rescue the inhibitory effect of XCT-790 (a known inverse agonist of ERRα) on breast cancer cell migration, presumably because XCT-90 mechanism of action involves degradation of ERRα [53,54]. Our finding that cholesterol mediates cell proliferation in an ERRα-dependent manner, was further confirmed using dose-competition assays between cholesterol and another well-characterized ERRα antagonist cpd29 [42,43,44] in MDA-MB-231 cells. Together, these results demonstrate that the knockdown of ERRα or inhibition of ERRα activity using XCT-790 or cpd29, results in abrogation of the enhancing effect of cholesterol on breast cancer cell proliferation and migration. These findings suggest that the stimulatory effects of cholesterol on cell proliferation and migration are mediated via ERRα.

Based on our findings, we propose that the mechanism by which cholesterol may exert its effects on ER+ and TNBC cells involves cholesterol binding to ERRα and changing its conformation, thereby enhancing ERRα interaction with its coactivator PGC-1α, with the increased ERRα-PGC-1α interaction resulting in augmented expression of ERRα itself (auto-induction) and of its target genes implicated in cellular metabolism, including IDH3A, PDK4, SOD2, GSTM1 and VEGF. We further propose that together, this may result in the reprogramming of tumor metabolism to provide sufficient biomass and detoxification against oxidative stress for breast cancer cells to proliferate and migrate faster (Figure 7). In contrast, treatment with the lipid soluble statin, lovastatin, an inhibitor of HMG-CoA reductase, the rate limiting enzyme in the cholesterol biosynthetic pathway, results in reduced cell proliferation and migration in breast cancer cells (Figure 7), likely via reducing cholesterol intracellular levels [39], and its effects are reversed by exogenous cholesterol addition. Together, our findings provide insight into the potential mechanisms underlying the increased risk of breast cancer associated with elevated levels of circulating cholesterol, and the protective effect of statins in improving breast cancer patients’ survival.

ERRα/PGC-1α/β activity is under the regulation of several oncogenic signals, including the PI3K/AKT/mTOR pathway which plays a key role in activating SREBP, a critical transcription factor involved in intracellular cholesterol synthesis in cancer cells [55]. Thus, cholesterol may provide a link between the mTOR pathway and ERRα/PGC-1 complex activation in cancer cells.

In the present study, we have not performed a detailed analysis of the cholesterol-ERRα binding complex and kinetics, by methods such as nuclear magnetic resonance (NMR), or X-ray crystallography. However, a previous study by another group has reported the binding kinetics and structural basis of cholesterol-ERRα interaction [29]. Using computational docking of cholesterol into the LBD of ERRα they have demonstrated that the hydroxyl group of cholesterol makes a hydrogen bond to E235 of ERRα’s LBD. Also, they have shown that F232 and L228 of ERRα possibly make important hydrophobic contacts with cholesterol [29]. A limitation of our study is that cholesterol lowering drugs like statins have been reported to have cholesterol-independent effects [40,41]. Unfortunately, there are no known cholesterol depleting agents that do not exhibit cholesterol-independent effects. While our findings in the current study show that cholesterol enhances cell proliferation and migration in an ERRα-dependent manner, the question as to whether other upstream targets are involved in this process remains to be determined. In the current study, we have used a cholesterol concentration of 5 μM for MDA-MB-231 and 10 μM for MCF-7 cells, because these doses showed the optimal response in the biological assays used. It is not possible to ascertain the physiologic relevance of the dose chosen as it is difficult to mimic in vivo concentrations of extracellular or intracellular cholesterol levels in vitro in the cell lines. In addition, it should be noted that the results presented in the present study are limited to a representative triple negative (MDA-MB-231) and to a representative ER+ (MCF-7) breast cancer cell line; using other breast cancer cells expressing varying levels of ERRα would strengthen the clinical implications of the current study.

## 5. Conclusions

In the present study, we demonstrate that cholesterol binds ERRα, enhances its interaction with its transcriptional co-activator PGC-1α and promotes ERRα transcriptional activity in ER-positive and in triple-negative breast cancer cells. Furthermore, we show that cholesterol activates several ERRα metabolic target genes and enhances cellular proliferation and migration, ERRα being required for these effects. Statins inhibit cell proliferation and migration in both ER-positive and triple-negative breast cancer cells, possibly by decreasing intracellular cholesterol levels [39]. Importantly, exogenous cholesterol is able to rescue these effects of statin.

There is increasing evidence that the expression levels of ERRα are higher in human breast tumors when compared to normal breast tissue [23], and that ERRα overexpression is associated with adverse clinical outcome and recurrence in breast cancer patients [52,56,57]. Thus, it has been suggested that the expression of ERRα could be used as a marker of unfavorable prognosis and response to therapy in breast cancer [56]. The interest in inhibiting ERRα activity in breast cancer patients is based on ERRα’s strong involvement in regulating a vast array of oncogenic functions, including metabolic reprograming of cancer cells [58,59]. Thus, the identification of cholesterol as an endogenous agonist of ERRα provides a potential avenue for targeting intracellular cholesterol action to globally impinge on the metabolic impairments in cancer cells. Further studies are warranted to explore the potential of drugs such as statins and SREBP inhibitors to prevent or treat breast cancer, in particular TNBC, which has a poor prognosis and no satisfactory treatment options. Furthermore, identification of cholesterol as an agonist of ERRα and a regulator of ERRα target gene expression, and proliferation in ER+ and TNBC cells, also has relevance to other subtypes of breast cancer, like the human epidermal growth factor receptor 2 positive (HER2+) subtype, and other cancer types such as prostate, ovary, and colorectal cancers, where ERRα is overexpressed and known to play a pathological role.

## Figures and Tables

**Figure 1 cells-09-01765-f001:**
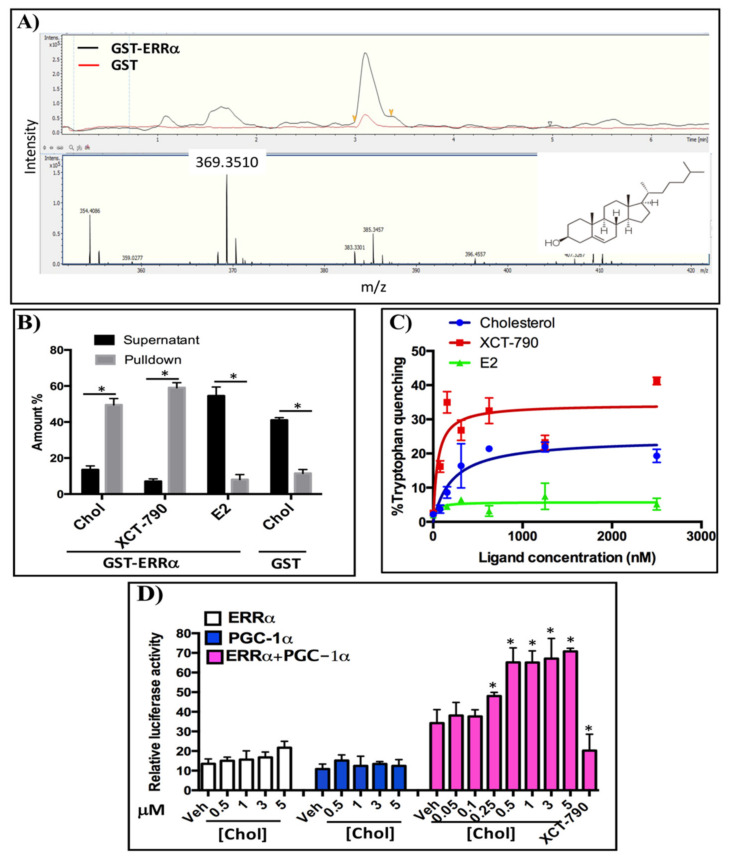
Cholesterol binds to ERRα and increases its activity: (**A**) Human pregnancy serum was incubated with Sepharose beads coupled to GST-ERRα-LBD and or unfused GST for 24 h at 4 °C and pull-down samples were analyzed using LC-MS. The upper panel shows the UV chromatogram and the lower panel displays the mass spectrum. The cholesterol structure is located at the right corner of this spectrum. (**B**) Cholesterol directly binds to ERRα-LBD. GST-ERRα-LBD pull down assays were performed and cholesterol concentrations were measured using LC-MS in MRM mode. XCT-790 and E2 concentrations were determined using a UV–vis spectrophotometer. Amounts are reported as % input. (**C**) Relative affinity of cholesterol for ERRα was assessed using a tryptophan quenching assay with fluorescence excitation at 295 nm and florescent emission at 310 nm. (**D**) Cholesterol increases transcriptional activity of ERRα in a PGC-1α dependent manner in a luciferase reporter assay. HEK 293 cells were transiently co-transfected with a pS2-LUC reporter plasmid with or without expression vectors for ERRα and the PGC-1α co-activator. The data are representative of 3 independent experiments. A value of *p* < 0.05 compared with the control group was considered significant (*).

**Figure 2 cells-09-01765-f002:**
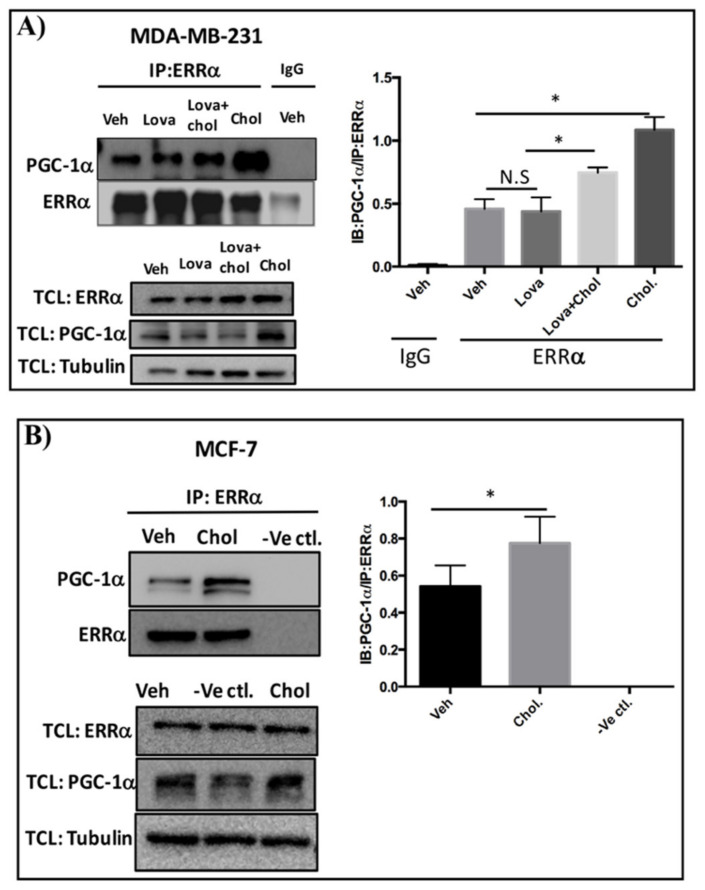
Cholesterol enhances ERRα-PGC-1α interaction in breast cancer cells. (**A**) MDA-MB-231 cells were treated with vehicle (Veh), lovastatin (Lova), lovastatin+cholesterol (Veh + Chol) or cholesterol (Chol) at 5 μM for 24 h. Cell lysates were subjected to immunoprecipitation (IP) with anti-ERRα or control IgG antibody. The protein complexes were separated by SDS-PAGE and immunoblotted (IB) with anti-PGC-1α or anti-ERRα antibodies to detect PGC-1α as a co-immunoprecipitated protein and ERRα as an immunoprecipitated protein. 2% of total cell lysate (TCL) were used to detect the endogenous levels of ERRα and PGC-1α. To quantify PGC-1α/ERRα ratio, densitometry analysis of PGC-1α and ERRα proteins derived from the same immunoblot was performed using ImageJ software. (**B**) MCF-7 cells were treated with vehicle (Veh) or cholesterol (Chol) at 10 μM for 24 h. Immunoprecipitation was performed using a Pierce co-immunoprecipitation (Co-IP) kit. The cell lysates were incubated with the AminoLink Plus Coupling Resin and immunoprecipitated with an ERRα antibody. Also as a negative control (-Ve Ctl.), the vehicle-treated cell lysate received the same concentration of anti-ERRα antibody except that the AminoLink Plus Coupling Resin was replaced with a Pierce Control Agarose Resin that is not amine-reactive, preventing ERRα antibody from binding to the resin. The bound proteins were eluted and separated by SDS-PAGE and immunoblotted with anti-PGC-1α or anti-ERRα antibodies to detect PGC-1α as a co-immunoprecipitated protein and ERRα as an immunoprecipitated protein, respectively. 2% of TCL were used to detect the endogenous protein levels of ERRα and PGC-1α. To quantify PGC-1α/ERRα ratio, densitometry analysis of PGC-1α and ERRα proteins derived from the same immunoblot was measured using ImageJ software. A minimum of 3 independent experiments were performed. A value of *p* < 0.05 was considered significant (*).

**Figure 3 cells-09-01765-f003:**
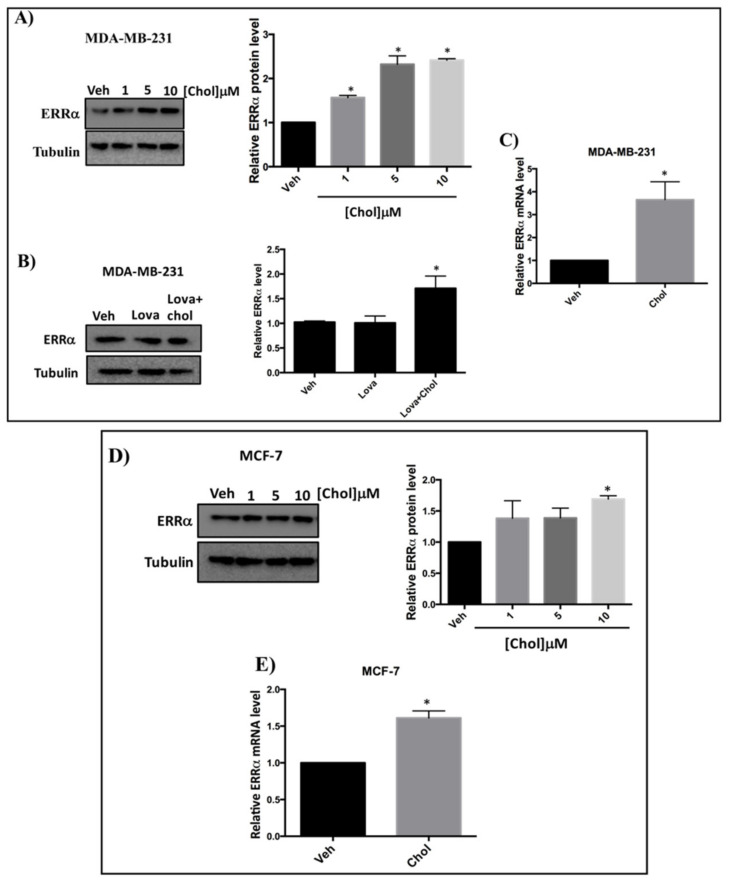
Cholesterol increases ERRα protein and mRNA levels in MDA-MB-231 and MCF-7 cells in a dose dependent manner. (**A**) Cholesterol increases ERRα protein levels in MDA-MB-231 cells in a dose dependent manner. MDA-MB-231 cells were treated with vehicle (Veh) and varying concentrations of cholesterol (1, 5, 10 μM) and subjected to western blotting. (**B**) Cholesterol increases ERRα protein levels even in the presence of lovastatin in MDA-MB-231 cells. MDA-MB-231 cells were treated with vehicle (Veh), lovastatin (Lova) or Lovastatin + cholesterol (Lova + Chol) at 5 μM for 24 h. Cell lysates were subjected to western blotting. Relative ERRα protein levels were assessed using ImageJ software. (**C**) Cholesterol induces ERRα mRNA levels in MDA-MB-231 cells. Cells were treated with 5 μM cholesterol for 24h and ERRα mRNA levels were assessed by RT-qPCR and were normalized to endogenous GAPDH. (**D**) Cholesterol increases ERRα protein levels in MCF-7 cells in a dose-dependent manner. MCF-7 cells were treated with varying concentrations of cholesterol (1, 5, 10 μM) or vehicle (Veh) and subjected to western blotting. Relative ERRα protein levels were calculated using ImageJ software. (**E**) Cholesterol induces ERRα mRNA levels in MCF-7 cells. Cells were treated with 10 μM cholesterol for 24 h and ERRα mRNA levels were assessed by RT-qPCR, and were normalized to endogenous GAPDH. A minimum of three independent experiments were performed. A value of *p* < 0.05 was considered significant (*).

**Figure 4 cells-09-01765-f004:**
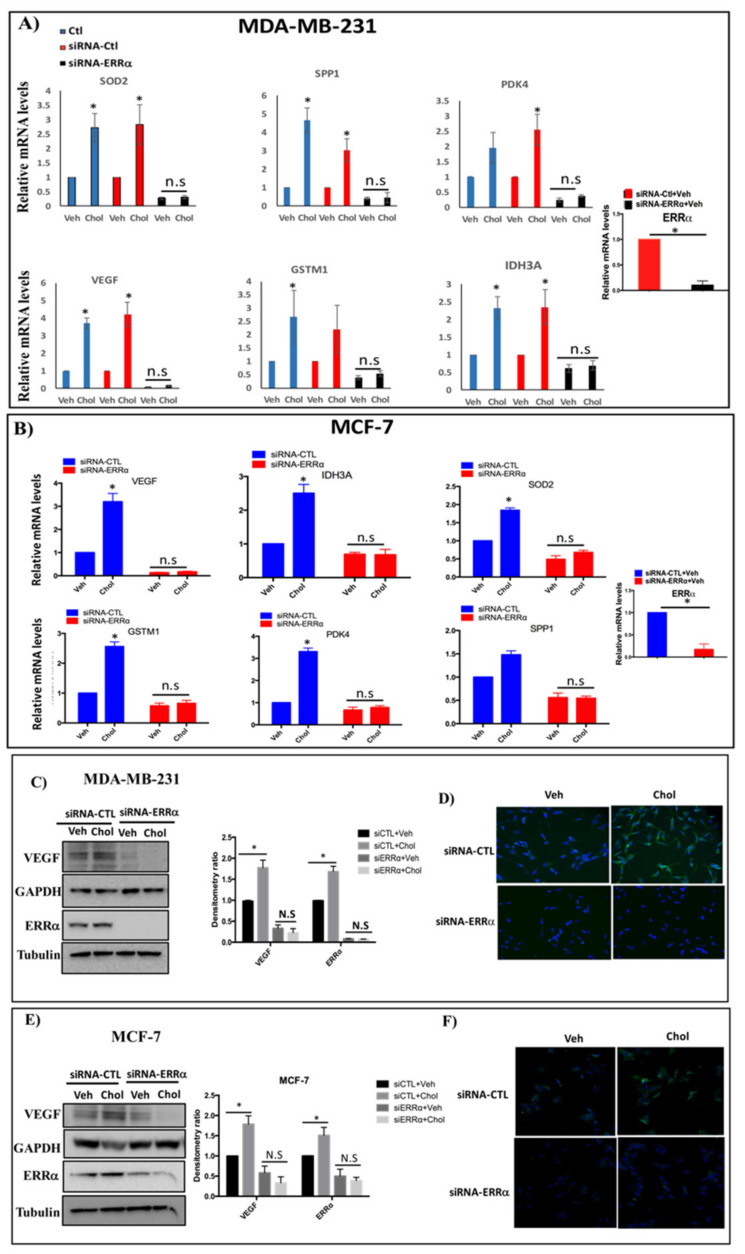
Cholesterol enhances expression of ERRα metabolic target genes through ERRα in MDA-MB-231 and MCF-7 cells. (**A**,**B**) Cells were transfected with siRNA-ERRα (siERRα) or siRNA-control (siRNA-CTL) for 48 h, and treated with vehicle (Veh) or cholesterol (Chol, 5 μM) for 24 h. Total RNA was extracted and analyzed by RT-qPCR. Genes detected included: isocitrate dehydrogenase 3A (IDH3A), pyruvate dehydrogenase kinase 4 (PDK4), vascular endothelial growth factor (VEGF), glutathione s-transfrase M1(GSTM1), superoxide dismutase 2 (SOD2) and secreted phosphoprotein 1(SPP1). The mRNA data were normalized to endogenous GAPDH (**C**,**E**) Cholesterol increases ERRα-induced VEGF protein expression in MDA-MB-231 and MCF-7 cells using western blotting. The blots for ERRα and VEGF were generated from the same cell lysates loaded on 2 different gels due to the similar molecular weights of the two proteins. GAPDH was used as a loading control for VEGF and tubulin was used as a loading control for ERRα. The densitometry ratio was calculated using ImageJ software. (**D**,**F**) Immunocytochemistry (ICC) was performed to detect VEGF protein expression using anti-VEGF antibody. All cells were transfected with siRNA-ERRα or siRNA-control (siRNA-CTL), and were treated with cholesterol (5 μM for MDA-MB-231, 10 μM for MCF-7 cells) or with vehicle for 24 h. DAPI is shown in blue and VEGF in green. A minimum of three independent experiments were performed. A value of *p* < 0.05 was considered significant in the comparison with the control group (*).

**Figure 5 cells-09-01765-f005:**
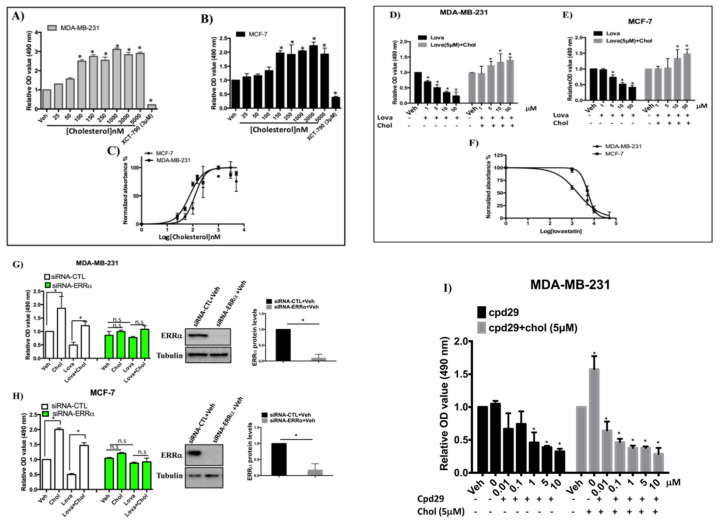
Cholesterol increases cell proliferation of breast cancer cells in an ERRα-dependent manner. (**A**,**B**) In order to obtain half maximal effective concentrations (EC_50s_) of cholesterol in MDA-MB-231 and MCF-7 cells, an MTS Cell Proliferation Assay kit was used to assay the cell proliferation of MDA-MB-231 and MCF-7 cells with varying concentrations of cholesterol (as indicated in the Figure) on day 5. (**C**) EC50s were calculated for both MDA-MB-231 and MCF-7 cells using the Prism software. (**D**,**E**) Cell proliferation in the presence of Lovastatin (Lova) and lovastatin (5 μM) + cholesterol (Chol), at varying concentrations indicated in the Figure, was measured using an MTS kit on day 5. (**F**) IC50s of lovastatin were calculated for both MDA-MB-231 and MCF-7 cells using Prism. (**G**,**H**) MDA-MB-231 and MCF-7 cells were transfected with siRNA-ERRα or siRNA-control (siRNA-CTL) and the cells were treated with vehicle, cholesterol (chol), lovastatin (lova) or lovastatin + cholesterol (lova + chol), all at 5 μM for MDA-MB-231 cells and at 10 μM for MCF-7 cells. Cell proliferation assays were performed using an MTS kit on day 5. Cell lysates were immunoblotted using an anti-ERRα antibody. A value of *p* < 0.05 was considered significant (*). (**I**) MDA-MB-231 cells were treated with compound 29 (cpd29) in varying concentrations as indicated in the figure (black bars). Also, cholesterol at a fixed concentration of 5 μM was co-administered with varying concentrations of cpd29 as indicated in the figure (gray bars); 0 indicates cholesterol (5 μM) alone, without cpd29 treatment. Cell proliferation assays were performed using an MTS kit on day 6. A minimum of three independent experiments were performed. All treatments were compared to the respective vehicle group, and a value of *p* < 0.05 was considered significant (*).

**Figure 6 cells-09-01765-f006:**
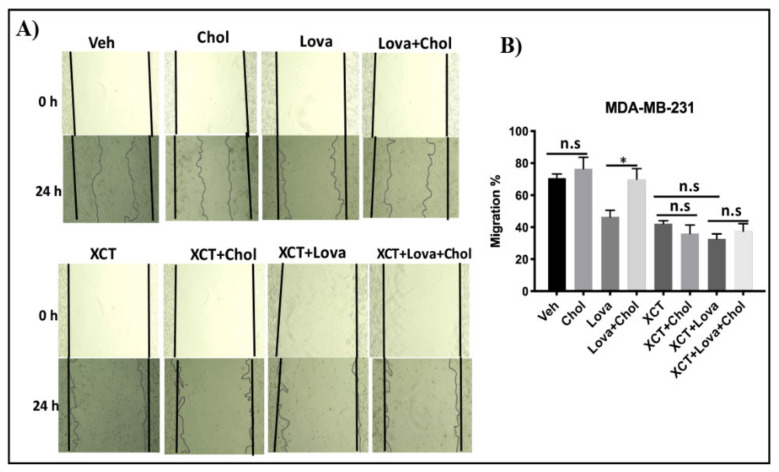
(**A**) Lovastatin decreases the migration of MDA-MB-231 cells, while cholesterol rescues the effect of lovastatin, but not the effect of XCT-790 using a scratch-wound migration assay, MDA-MB-231 cells were treated with vehicle(Veh), cholesterol (chol), lovastatin (Lova), Lovastatin + cholesterol (Lova + Chol), XCT-790 (XCT), XCT-790 + cholesterol (XCT + chol), XCT-790 + lovastatin (XCT + Lova) or XCT-790 + lovastatin + cholesterol (XCT + Lova + Chol), all at 5 μM. Wound closure was monitored at 0 and 24 h, and representative images are provided. (**B**) Migration percentages were calculated as follows: migration % = (*T* 0 h scratch width − *T* 24 h scratch width/*T* 0 h scratch width) × 100. The results represent 3 independent experiments. A value of *p* < 0.05 was considered significant (*).

**Figure 7 cells-09-01765-f007:**
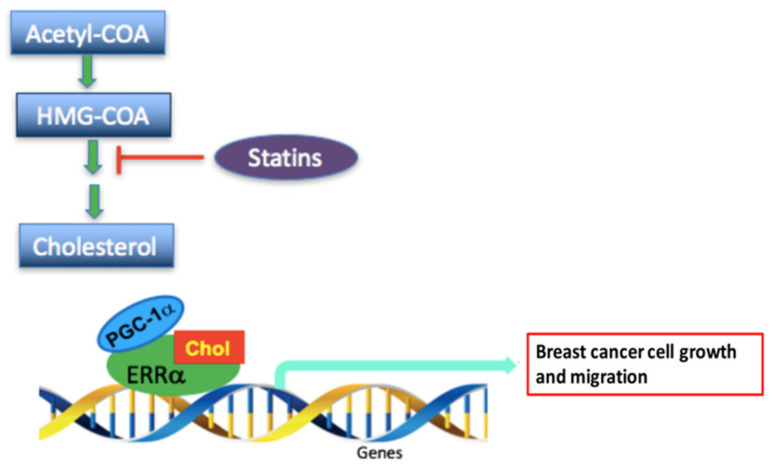
Schematic representation of the proposed mechanism by which cholesterol promotes breast cancer cell growth and migration via the ERRα-PGC-1α pathway and how statins (cholesterol lowering drug) may inhibit this effect. The proposed scheme depicts that cholesterol binds to ERRα and changes its conformation, which causes an increase in recruitment of PGC-1α and as a result induces transcription of the metabolic target genes of ERRα and increases breast cancer cell growth and migration. However, statins, drugs tthat inhibit HMG-CoA reductase, possibly lower cholesterol levels and as a result decrease breast cancer cell progression.

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
