# Peer review of "Cholesterol as an Endogenous Ligand of ERRα Promotes ERRα-Mediated Cellular Proliferation and Metabolic Target Gene Expression in Breast Cancer Cells"

_cells, 2020, doi:10.3390/cells9081765_

Round 1

Reviewer 1 Report

This manuscript (Cholesterol as an Endogenous Ligand of ERR Promotes ERR-Mediated Cellular Proliferation and Metabolic Target Gene Expression in Breast Cancer Cells) was written by Faegheh Ghanbari et. al.  I have the following questions:    

1. ERR is considered not to be regulated by E2. Why does Fig1B and C show that E2 will affect ERR?    

2. ERR is currently considered to be related to ERα to a considerable extent, but why hasn't relevant experiments and discussions been conducted?    

3. Both Fig2 and Fig3 MCF7 experiments lack Lova and Chol related experiments. I think the data of this experiment is important.    

4. The siRNA effect of ERRα in MCF7 is not significant, so the data of MCF7 in Fig4 is hard to believe.    

5. The appearance of VEGF in MCF7 in Fig4E is inconsistent with the appearance of ERR and Tubulin. Are they from the results of different experiments?    

6. In Fig 3 E, the Y-axis text and numbers overlap. Please correct it.

Reviewer 2 Report

This article describes the role of cholesterol as an endogenous ligand to the ERRα nuclear receptor, which is known to influence breast cancer progression. The authors use serum isolated from pregnant humans to screen for endogenous compounds that bind to recombinant ERRa – the compounds being identified by mass spectroscopy. In this manner, cholesterol was identified as being able to associate with ERRa. Confirmatory studies included GST pull down assays between cholesterol and ERRa. Cholesterol did not enhance ERRa transcriptional activity unless ERRa and PGC1a were being overexpressed. Certain ERRa target genes were induced by cholesterol. Cholesterol increased cellular proliferation, and the authors provide some evidence that this may be mediated through ERRa. While the data presented here are intriguing, the paper is a bit superficial, and the conclusions over-reach the data. In addition, there are several major concerns.

Strengths

  • This paper builds on previous reports, providing additional evidence that cholesterol binds ERRa.

Major concerns:

  • Important controls and thorough pharmacologic studies are lacking. For example: XCT790 is not used in co-treatment with cholesterol. A thourough dose-competition assay between XCT790 and cholesterol would be required to conclude that the effects of cholesterol (eg. on proliferation) are being mediated via ERRa.
  • Cholesterol alone in some assays (e.g. reporter and migration assays) did not induce biological activity, although it rescued lovastatin effects – but not those of XCT790. This would argue that ERRa and basal cholesterol are both important for migration, but on separate pathways.
  • 1D: data should not be normalized between groups. It is important to see whether ERRa or PGC1a expression increase reporter activity.
  • Figure 5D: a vehicle only control is lacking. X-axis: it is very unclear what is being manipulated – cholesterol, lovastatin or both?
  • A positive control post-siRNA should be included for Fig. 5G and H. For example, for H, does E2 still promote proliferation? This is important, as ERRa may just be required for proliferation, and not mediating the effects of cholesterol.
  • ERRa knockdown efficiency should be shown
  • siRNA against PGC1a should be used to rule out cholesterol interacting with targets upstream of ERRa (like PGC1a itself)
  • Figure 1B: Why would E2 decrease in pulldown?
  • Reliance on lovastatin. Other methods of cholesterol depletion should be used to complement results, adding confidence that the effects are not off-target (example: cyclodextrin, different class of statin, siRNA against HMGCR etc.).
  • Doses of cholesterol change between experiments, without appropriate rationale. The authors should also comment on the physiologic relevance of the doses chosen.
  • Discussion of previous work in the field is lacking.
  • Acknowledgment of caveats of data should be included. Without a crystal structure or other methods, it cannot be concluded that cholesterol binds in a traditional sense of ligand-NR binding.
  • Figure 7 is very premature, and should not be included. Although certain ERRa target genes (and not cellular processes such as TCA cycle) have been shown to be regulated by cholesterol, whether ERRa mediates this changes is not supported by the current data.
  • Only one representative cell line for each subtype (MDA MB 231 or MCF7) was used. Additional cell lines should be used to confirm the authors’ results, thereby increasing the rigor of their studies.

Minor concerns:

  • More information should be provided on the patients
  • Line 10: breast cancer is the 2nd leading cause of cancer-related death among women.
  • Line 76: missed: degrees Celsius.
  • Line 116-117: “Montreal university” – it should be University of Montreal?
  • Line 130: “Santa cruze” – name of the city – please provide full company name
  • Line 429: data that is not statistically significant cannot be interpreted as an increase.
  • Please consider reformatting figures, so that data are more readily seen and read.
  • Line 492- typing error in spelling of ‘gene’
  • Line 535-grammatical error in sentence
  • Line 555- FDA approved drugs do exist for TNBC.
  • Formatting for references is inconsistent.

Round 2

Reviewer 1 Report

The author has replied to my question, and I think it is a reliable article.